# Application of Dynamic Contrast-Enhanced Ultrasound in Evaluation the Activity of Crohn’s Disease

**DOI:** 10.3390/diagnostics14070672

**Published:** 2024-03-22

**Authors:** Ying Wang, Li Wei, Wen-Song Ge, You-Rong Duan, Wen-Jun Ding, Xiu-Yun Lu, Yun-Lin Huang, Sheng Chen, Yi Dong, Peng Du

**Affiliations:** 1Department of Ultrasound, Xinhua Hospital Affiliated to Shanghai Jiaotong University School of Medicine, Shanghai 200092, China; yunxi2009@126.com (Y.W.); weilii2022@163.com (L.W.); 21211320004@m.fudan.edu.cn (X.-Y.L.); 22111210067@m.fudan.edu.cn (Y.-L.H.); 21211210033@m.fudan.edu.cn (S.C.); 2Department of Gastroenterology, Xinhua Hospital Affiliated to Shanghai Jiaotong University School of Medicine, Shanghai 200092, China; gewensong@xinhuamed.com.cn; 3State Key Laboratory of Systems Medicine for Cancer, Shanghai Cancer Institute, Renji Hospital, Shanghai Jiao Tong University School of Medicine, 2200/25 Xietu Rd., Shanghai 200032, China; yrduan@shsci.org; 4Department of Anorectal Surgery, Xinhua Hospital Affiliated to Shanghai Jiaotong University School of Medicine, Shanghai 200092, China; wow_dingwenjun@163.com

**Keywords:** dynamic contrast-enhanced ultrasound (DCE-US), Crohn’s disease, activity, time–intensity curves (TICs), quantitative parameters

## Abstract

Background and Objective: The dynamic assessment of disease activity during the follow-up of patients with Crohn’s disease (CD) remains a significant challenge. In this study, we aimed to identify the role of dynamic contrast-enhanced ultrasound (DCE-US) in the evaluation of activity of CD. Methods: In the retrospective study, patients diagnosed with CD in our hospital were included. All the diagnoses were confirmed by clinical symptoms and ileocolonoscopical results. All patients underwent intestinal ultrasound and contrast-enhanced ultrasound (CEUS) examinations within 1 week of the ileocolonoscopy examinations. Acuson Sequoia (Siemens Healthineers, Mountain View, CA, USA) and Resona R9 Elite (Mindray Medical Systems, China) with curved array and Line array transducers were used. The CEUS examination was performed with SonoVue (Bracco SpA, Milan, Italy). DCE-US analysis was performed by UltraOffice (version: 0.3-2010, Mindray Medical Systems, China) software. Two regions of interest (ROIs) were set in the anterior section of the infected bowel wall and its surrounding normal bowel wall 2 cm distant from the inflamed area. Time–intensity curves (TICs) were generated and quantitative perfusion parameters were obtained after curve fittings. The Simple Endoscopic Score for Crohn’s disease (SES-CD) was regarded as the reference standard to evaluate the activity of CD. The receiver operating characteristic curve (ROC) analyses were used to determine the diagnostic efficiency of DCE-US quantitative parameters. Results: From March 2023 to November 2023, 52 CD patients were included. According to SES-CD score, all patients were divided into active group with the SES-CD score > 5 (n = 39) and inactive group SES-CD score < 5 (n = 13). Most of the active CD patients showed bowel wall thickness (BWT) > 4.2 mm (97.4%, 38/39) or mesenteric fat hypertrophy (MFH) on intestinal ultrasound (US) scan (69.2%, 27/39). Color Doppler signal of the bowel wall mostly showed spotty or short striped blood flow signal in active CD patients (56.4%, 22/39). According to CEUS enhancement patterns, most active CD patients showed a complete hyperenhancement of the entire intestinal wall (61.5%, 24/39). The TICs of active CD showed an earlier enhancement, higher peak intensity, and faster decline. Among all CEUS quantitative parameters, amplitude-derived parameters peak enhancement (PE), wash-in area under the curve (WiAUC), wash-in rate (WiR), wash-in perfusion index (WiPI), and wash-out rate (WoR) were significantly higher in active CD than in inactive CD (*p* < 0.05). The combined AUROC of intestinal ultrasound features and DCE-US quantitative perfusion parameters in the diagnosis of active CD was 0.987, with 97.4% sensitivity, 100% specificity, and 98.1% accuracy. Conclusions: DCE-US with quantitative perfusion parameters is a potential useful noninvasive imaging method to evaluate the activity of Crohn’s disease.

## 1. Introduction

Crohn’s disease (CD) is a chronic non-specific inflammatory bowel disease (IBD) of the digestive tract characterized by episodes of relapse alternating with periods of remission [1]. The incidence and prevalence of IBD in China have exhibited a significant upward trend over the past three decades. The burden of IBD is expected to grow continuously in China [2]. The early and accurate diagnosis is of vital importance for the clinical management of IBD. According to the 2018 European Crohn’s and Colitis Organization and the European Society of Gastrointestinal and Abdominal Radiology (ECCO-ESGAR) guidelines, it could be divided into active and inactive CD depending on the clinical, endoscopic, or biochemical evidence [3]. Active CD is characterized by transmural penetration/active inflammation which might lead to various complications, such as fistulas, fibrotic strictures, and abscesses [4,5]. A non-invasive and dynamic assessment of CD activity is essential for accurate clinical treatment and following up [6].

Currently, ileocolonoscopy is the first-line imaging examination for the diagnosis, monitoring, and management of CD [3]. According to the 2018 ECCO-ESGAR guidelines, endoscopic scoring systems including Crohn’s disease endoscopic index of severity (CDEIS) and the Simple Endoscopic Score for Crohn’s disease (SES-CD) are most commonly used to quantify the inflammation activity of CD [3,7]. Among them, the SES-CD is a simple, rapid endoscopic scoring system to be used for objectively assessing mucosal lesions in CD [8]. SES-CD mainly includes four variables including ulcer size, the degree of an ulcerated surface, the range of affected surface, and bowel stenosis. SES-CD was equal to sum the scores for the four variables in all explored segments, including ileum, right colon, transverse colon, left colon, and rectum. Histopathologically, CD is a transmural process often accompanied by fistulas, abscesses, and other complications and those transmural changes or microvascular perfusion changes of bowel walls cannot be evaluated by ileocolonoscopy. Mucosal healing does not mean whole-layer healing. Also, ileocolonoscopy is an invasive procedure and poorly tolerated by some patients with a risk of perforation [7,9]. A non-invasive and quick imaging method is necessary in the clinical follow-up of the treatment response of CD.

Currently, different imaging techniques, including magnetic resonance enterography (MRE), computed tomography enterography (CTE), and intestinal ultrasound (US) have been used for the evaluation of CD activity. Studies had shown that the MRE had 69–92% sensitivity and 41–64% specificity for evaluating the activity of CD [10]. CTE had 77.5% sensitivity and 90.8% specificity for evaluating the activity of CD [11]. Intestinal US has the advantage of repeatable, radiation-free, well tolerated, less expensive, and general availability [12]. Previous studies had shown that the sensitivity and specificity of US features combining of BWT and color Doppler signal in evaluation of CD activity was reported to be 86.6–97.3% and 63.6–87%, respectively [10]. According to 2018 European Federation of Societies for Ultrasound in Medicine and Biology (EFSUMB) guidelines, intestinal US was recommended to assess CD location and activity. The color Doppler technique can assist in the semi-quantitative evaluation of intestinal wall vascularity that is useful to assess CD activity [13].

More recently, contrast-enhanced ultrasound (CEUS) has been used for quantifying the microvascular perfusion [14]. It has been shown to be a sensitive imaging method for evaluating microvascular perfusion of liver lesions, pancreatic lesions, and renal lesions [15,16]. Dynamic contrast-enhanced US (DCE-US) is a quantitative imaging technique, by generating TICs and perfusion quantitative parameters [17], it has been proven to be a repetitive and objective evaluation methods for the diagnosis and follow-up of treatment responses [18].

The aim of this study was to investigate the added value of DCE-US in quantitatively assessing activity of CD.

## 2. Patients and Methods

This retrospective study was approved by the hospital ethics committee (ID No. XHEC-C-2023-028-1). The requirement for informed consent was waived. The followed procedure was in accordance with the principles outlined in the declaration of Declaration of Helsinki.

### 2.1. Patients

Patients diagnosed with CD, with confirmed clinical symptoms and ileocolonoscopical results in our hospital, were included and retrospectively analyzed. All patients underwent intestinal US and CEUS examinations within 1 week of the ileocolonoscopy examinations. The inclusion criteria of this study were as follows: (1) Patients aged from 18 to 85 years; (2) The patients underwent intestinal US, CEUS, and ileocolonoscopy; (3) The activity of CD was scored according to SES-CD criteria; (4) Bowel walls could be clearly visible on conventional intestinal US; (5) All patients had available clips of Digital Imaging and Communications in Medicine (DICOM) format at least 2 min. The exclusion criteria of this study were as follows: (1) Patients had a known allergic history of sulfur hexafluoride microbubbles; (2) Patients with incomplete imaging data; (3) Patients had previous history of intestinal segments surgical resection.

### 2.2. Ultrasound Examination Technique

A series of standard intestinal US and CEUS examinations for patients were performed by two experienced radiologists (independently with more than 20 years and 10 years of CEUS examination of intestinal ultrasound). Two advanced ultrasound devices were used including Acuson Sequoia (Siemens Healthineers, Mountain View, CA, USA; 5C1 convex array probe, 10L4 linear array probe) and Resona R9 Elite (Mindray Medical Systems, China; SC6-1U convex array probe, L15-3WU linear array probe).

#### 2.2.1. Intestinal Ultrasound Examination

Patients fasted for 8 h and were given a liquid diet 1 day before intestinal B mode ultrasound (BMUS) examination.

First, the ileocecal section, ascending colon, transverse colon, descending colon, sigmoid colon, and rectum were scanned with a convex array probe (Figure 1a). Then, a comprehensive ultrasound scan procedure from the upper left abdomen to the lower right abdomen was performed for the small intestine (Figure 1b) [19]. Afterwards, the thickest intestinal segment was observed using the linear array probe.

The imaging features observed by intestinal US include BWT (mm), color Doppler signal of bowel wall, existence of mesenteric lymph nodes (MLN), existence of mesenteric fat hypertrophy (MFH), and the presence of abdominal complications in CD (bowel stenosis, formation of fistula, and abscess). BWT was obtained when the calipers were placed between the serosa and superficial mucosal interface in the longitudinal direction (thickened bowel wall with BWT > 3 mm) [13]. Color Doppler signal of bowel wall was observed at the most thickened segments by power Doppler imaging and estimated according to Limberg score [20], which was classified as follows: Grade 0, normal intestinal wall; Grade I, thickened bowel wall with no blood flow signals; Grade II, thickened bowel wall with dot- or short-like blood flow signals; Grade III, thickened bowel wall with longer blood flow signals; Grade IV, thickened bowel wall with longer blood flow signals connected to the mesentery. MLN was classified as normal and enlarged MLN (elongated with a lesser diameter > 5 mm). MFH was defined as a hyperechoic tissue encircling the diseased intestinal segment [13].

#### 2.2.2. Contrast-Enhanced Ultrasound Examination

The longitudinal section of the thickest intestinal segment was observed on the CEUS using linear array probe (10L4 with a frequency of 3–10 MHz and L15-3WU with a frequency of 3.8–15.4 MHz). Then, 2.4 mL of SonoVue (Bracco SpA, Milan, Italy) was dissolved in 5.0 mL saline, mixed with shaking, and then injected through antecubital venous with a bolus fashion for CEUS examination. All dynamic images of CEUS were observed for at least 2 min. All ultrasound images were digitally recorded in DICOM (Digital Imaging and Communications in Medicine) format for further analysis.

CEUS enhancement patterns were classified four categories [21]: pattern I, complete enhancement of the bowel wall from the mucous to serous layer; pattern II, the enhancement of the bowel wall including mucous, submucosa and muscularis mucosae; pattern III, enhancement only in submucosa layer; pattern IV, absence of enhancement in the entire bowel section.

The CEUS DICOM cine loops were dynamically observed by UltraOffice (version: 0.3-2010, Mindray Medical Systems, China) software. Time–intensity curves (TICs) were fitted and linearized based on manually drawn regions of interest (ROIs) at the anterior section of the inflammation bowel wall and the normal bowel wall.

After curve fitting, the result was considered reliable when the quality value of fit was >75%. The quantitative parameters of DCE-US were recorded, including wash-in rate (WiR, the maximum slope of the TIC in terms of the tangent line to the ascending part of the curve), peak enhancement (PE), wash-out rate (WoR, the minimum slope of the curve, in terms of the tangent line to the descending of the curve), wash-in area under the curve (WiAUC, area under the TIC from arrival time to the peak enhancement), wash-out AUC (WoAUC, the area under the TIC from the peak enhancement to the end of the curve), wash-in and wash-out AUC (WiWoAUC, the area under the TIC), rise time (RT), time to peak (TTP), mean transit time (mTT), fall time (FT), wash-in perfusion index (WiPI, WiAUC/RT) [17].

### 2.3. Ileocolonoscopy Evaluation

After the preparation of standard intestinal cleaning with 4 L of polyethylene glycol electrolyte solution, the ileocolonoscopy was performed by endoscopists who have had more than 10 years of experience in inflammatory bowel disease diagnosis. The Simple Endoscopy Score for CD (SES-CD) was used to evaluate the mucosal inflammatory activity in CD [8]. The four evaluating variables were as follows: size of the ulcer, degree of ulcerated surface, range of affected surface, and the existence of stenosis. Each parameter was scored from 0 to 3 in each section of the following segments: ileum, right colon, transverse colon, left colon and rectum, with a maximal total SES-CD score of 48. When the SES-CD score > 5, it was considered endoscopic activity [8,22].

### 2.4. Statistical Analysis

Statistical analyses were performed with SPSS software for Windows (version 26.0; SPSS, Chicago, IL, USA) and GraphPad Prism 8 (GraphPad Software Inc., San Diego, CA, USA). Continuous data consistent with a normal distribution were compared by an independent two-sample *t* test. Continuous variables of abnormal distribution were compared by Mann–Whitney test. Chi-square test was used for analyzing categorical variables. The cutoff value was obtained from the receiver operating characteristic (ROC) curve when the Youden index was maximum, which is the same as the sensitivity and specificity. The diagnostic performance was assessed by ROC curve analysis according to SES-CD. The difference was statistically significant when *p* < 0.05.

## 3. Results

### 3.1. Patients’ Characteristics

From March 2023 to November 2023, fifty-two CD patients were retrospectively included. According to their SES-CD results, patients were divided into two groups: active CD group with SES-CD > 5 (n = 39) and inactive CD group with SES-CD ≤ 5 (n = 13). While comparing between two groups, the values of c-reactive protein (CRP) and erythrocyte sedimentation rate (ESR) are higher in active CD group (*p* < 0.05) [Table 1].

### 3.2. Intestinal Ultrasound Features

On the BMUS scan, BWT in the active CD group was thicker than that in the inactive CD (6.5 mm ± 1.7 vs. 3.2 mm ± 1.0, *p* < 0.001). Most patients in the active CD group showed an MFH feature (69.2%, 27/39). According to the Limberg score criteria, the color Doppler signal of the bowel wall more commonly showed Limberg II with dot- or short-like blood flow signals in active CD patients (56.4%, 22/39), while Limberg 0 showed a normal intestinal wall (46.1%, 6/13) in inactive CD patients. The CEUS enhancement patterns of the bowel segment with active inflammation were mainly Pattern I (Pattern I: 61.5%, 24/39), which showed the complete hyperenhancement of the entire intestinal wall from the mucous to serous layer. However, the CEUS enhancement patterns in the inactive CD group were mainly Pattern III (61.5%, 8/13) with hyperenhancement only in the submucosa layer [Table 2, Figure 2 and Figure 3].

### 3.3. Dynamic Contrast-Enhanced Ultrasound (DCE-US) Quantitative Analysis

TICs of CD patients were created on the CEUS DICOM images of 2 min. When compared with those of the bowel wall of inactive CD patients, the TICs of the active inflammation segment in active CD revealed earlier hyperenhancement, higher peak intensity, and faster decline [Figure 2 and Figure 4].

After the curve fitting of the TICs, various quantitative parameters of DCE-US were generated with the fit quality exceeding 75%. Quantitative parameters including RT, TTP, and FT were significantly lower in the inflammation segment in the active CD group than in the inactive CD group (*p* < 0.05). Parameters including PE, WiAUC, WiR, WiPI, and WoR were significantly higher in the active CD group than in the inactive CD group (*p* < 0.05) [Figure 4].

### 3.4. Diagnostic Efficiency of Intestinal Ultrasound Features and Perfusion Quantitative Parameters

The ROC curve analysis of positive intestinal ultrasound features and perfusion quantitative parameters was performed. While using BWT > 4.2 mm as a cut-off value, this showed 94.6% of sensitivity, 97.4% of specificity, and 0.946 of AUROC in the assessment of CD activity. The combined AUROC of the positive CEUS perfusion quantitative parameters intestinal US features (including BWT, color Doppler signal, existence of MFH) and CEUS enhancement pattern was 0.987 (95% CI: 0.958–1.000), with 97.4% sensitivity, 100% specificity, 98.1% accuracy, which is significantly higher than that of BWT [Figure 5].

## 4. Discussion

As a transmural inflammation, half of the activity of CD patients shows various intestinal complications, such as strictures, fistulas, and abscesses, etc. [1], which need further clinical treatments, including surgery, etc. Therefore, the early and non-invasive evaluation of CD activity may prevent the need for further surgical interventions. Because of its unique advantages, in terms of enabling real-time, radiation-free, and non-invasive imaging, ultrasound is an optimal imaging methods for the clinical follow-up and evaluation of inflammation activity in CD patients. According to the EFSUMB guidelines, intestinal US is recommended for assessing CD activity [13].

BMUS is a first-line clinical imaging methods. In our results, BWT showed a satisfactory diagnostic efficiency for evaluating the inflammatory activity of CD (AUROC: 0.946). The BWT cutoff value in our study for distinguishing the active and inactive CD was 4.2, mm which is similar to a meta-analysis [23]. The BWT is related to the clinical and biochemical activity of CD. The thickening of the bowel wall with good repeatability is the simple and convenient method for evaluating the activity of CD patients [24,25].

Previously, it has been reported that MFH is another common feature of active CD. It was found by intestinal US in approximately 40–50% of CD patients with the reported sensitivity and specificity > 83% [13]. Among the active CD patients in our cohort, 69.2% showed an MFH feature and none of the inactive CD patients showed an MFH feature. Although MFH seems to be related to transmural inflammation, fibrosis, stricture, etc., the role it plays in the onset and development of CD is not fully understood [13,26]. The existence of MFH is related to clinical and biochemical disease activity, which may disappear or improve in CD patients who have responded to medical treatment [13].

Increased angiogenic activity occurs in CD, resulting in increased microvascular blood perfusion, which could be evaluated by color flow imaging method [21], which increased the color Doppler signals and could identify active CD with a sensitivity of 95% and a specificity of 66.6% [27]. Our study showed that color flow imaging could estimate the CD activity with a sensitivity of 87.1% and a specificity of 69.2%. However, the color flow imaging has some limitations such as the detection of low-velocity blood flow in small blood vessels in the intestinal wall and blood flow in the deep bowel. In our cohort, the active inflammation of the intestinal wall of some patients showed a spotty blood flow signal on color flow imaging, but pattern I showed a complete enhancement of the intestinal wall on CEUS.

By the injection of the CEUS contrast agents, CEUS could also improve the detection of hypervascularity and perfusion in the capillaries and in the deep bowel [13]. CEUS can visualize the microvascular perfusion of the active inflammation bowel wall directly [28]. In a meta-analysis of eight studies involving 428 patients, the overall pooled diagnostic accuracy for CEUS in differentiating active CD exceeded 80% [29]. The CEUS of the bowel wall segment with active inflammation mostly showed enhancement pattern I (complete enhancement of the intestinal wall from the mucous to serous layer) and pattern II (enhancement of the intestinal wall including mucous, muscularis mucosae, and submucosa) [21].

DCE-US is a technique to quantify the tissue perfusion down to the capillary level based on phase-specific enhancement after injecting the contrast agents for diagnostic ultrasound, which can be applied to the evaluation of inflammatory activity. Since it is assumed that the signal intensity in DCE-US is proportional to the number of microbubbles, the TIC parameters are correlated with the vascularization of the analyzed region [30]. After curve fitting, quantitative parameters from TICs provide an objective assessment of the time and degree of wash-in and wash-out to quantitatively evaluate the bowel micro-vascular perfusion in active and inactive CD patients. A previous study showed that the multi-class ROC analyses of PE could separate the mild, moderate, and severe disease of CD [23]. A significant result in our study was that the WiAUC, WiR, PE, WiPI, and WoR of DCE-US in active CD were higher than those in inactive CD. The RT, TTP, and FT were significantly lower in the active group than in the inactive group. The quick wash-in followed by the rapid and early wash-out was the typical enhancement pattern of active CD patients, which was different from that of the inactive CD group [31]. An increased blood supply of CD with vascular proliferation, expansion, and distortion might explain the result [32].

Previously, the diagnostic performance of US based on BWT was 90% sensitivity and 93% specificity [33]. In our study, all positive perfusion parameters and intestinal US features (including BWT, color Doppler signal, and MLN) were combined, and the best diagnostic performance was obtained with AUROC 0.987 (95% CI: 0.958–1.000), with 97.4% sensitivity, 100% specificity, and 98.1% accuracy. Repetitive perfusion parameters obtained by DCE-US evaluate the degree of inflammation, which can be more accessible and explained by the operator, regardless of the level of experience [30]. Based on DCE-US quantitative analysis, a relevant ultrasound diagnostic model can be constructed to objectively evaluate the CD activity in future studies.

Our study has several limitations. Firstly, in consequence of the retrospective study, some bias might be present, and thus, a prospective research design is needed. Secondly, it is a single center experience with a relatively small sample size, and further multi-center studies with large case series are needed to verify our results.

## 5. Conclusions

In summary, DCE-US with quantitative perfusion parameters is a potentially useful non-invasive imaging method for evaluating the activity of Crohn’s disease. PE, WiAUC, WiR, WiPI, WoR, RT, TTP, and FT may be valuable quantitative parameters.

## Figures and Tables

**Figure 1 diagnostics-14-00672-f001:**
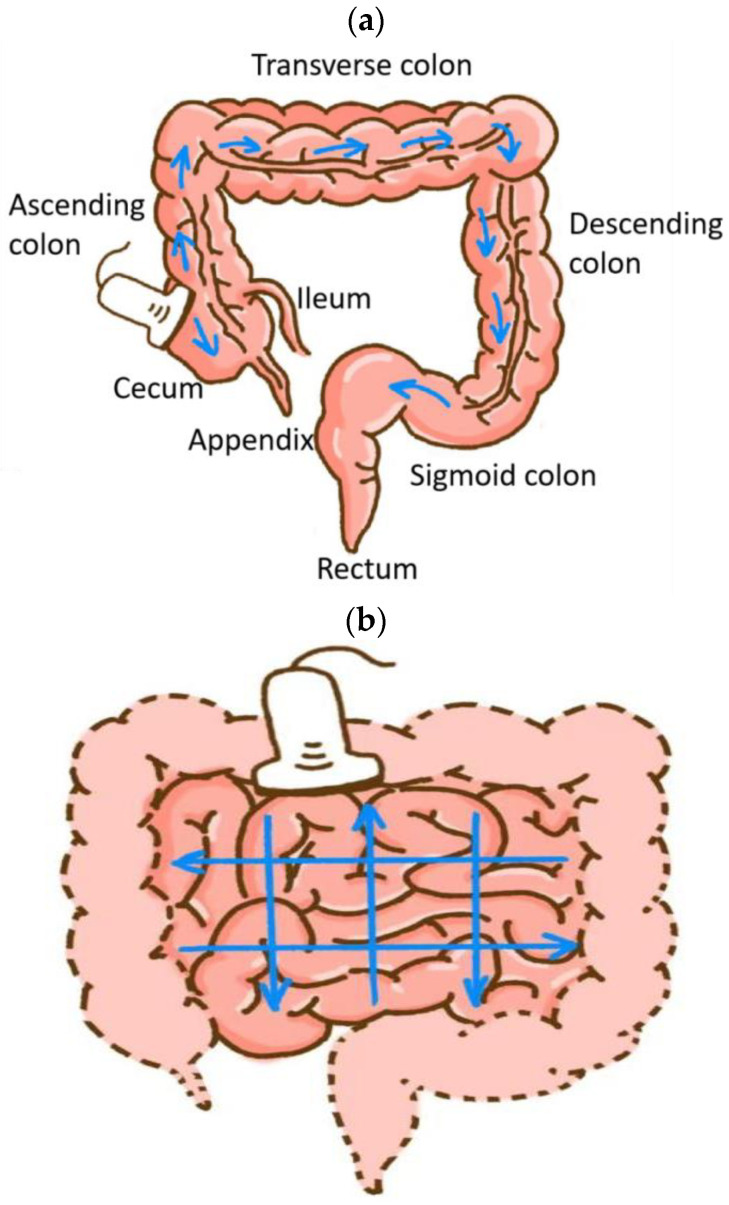
Schematic representation of comprehensive ultrasound scan procedure for the colon. Blue arrows indicate the sweeping order (**a**); Schematic representation of comprehensive ultrasound scan procedure for the jejunum and the ileum. Blue arrows indicate the parallel shifts of the probe from two directions (**b**).

**Figure 2 diagnostics-14-00672-f002:**
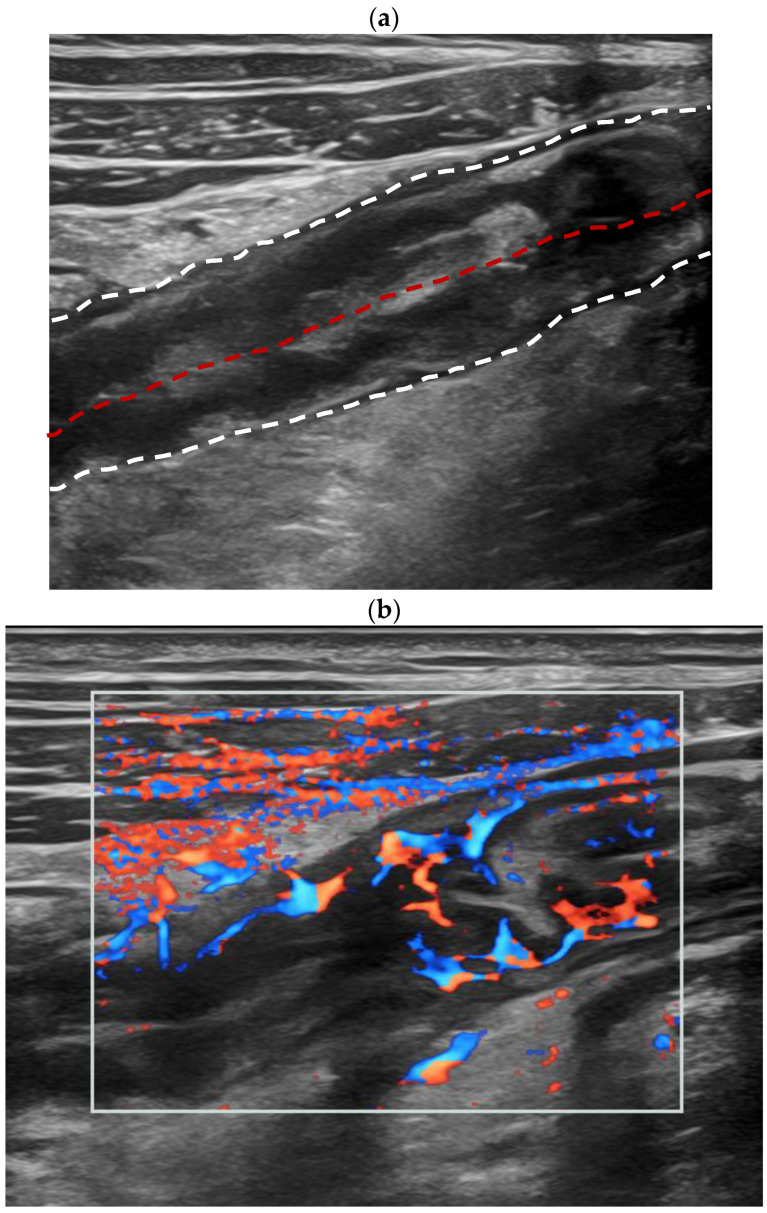
A twenty-year-old female patient diagnosed with active Crohn’s disease. Longitudinal scan of intestinal ultrasound showed the bowel wall layer is indistinctively stratified, thickness of the ileum bowel wall is 6.0 mm (the white dotted lines display the boundary between the serosal layer of the bowel wall and the surrounding tissue, and the red dotted lines display the bowel lumen, and the same is applied to the figure legends shown below) (**a**); Color Doppler signal of the active inflammation bowel wall is Limberg IV, longer blood flow signals connected to the mesentery could be detected in thickened intestinal wall (**b**); Contrast-enhanced ultrasound (CEUS) of the bowel wall segment with active inflammation showed complete enhancement of the bowel wall from the mucous to serous layer (enhancement pattern I, (**c**)). Two regions of interest (ROIs) were set inside the thickened bowel wall (orange ROI) and in the normal bowel wall (green ROI) as a reference region. Compared with the normal bowel (green curve), the TIC of the active inflammation bowel wall (orange curve) revealed an earlier and quicker hyperenhancement during the arterial phase, showed a higher peak intensity and a faster decline. After the dynamic contrast-enhanced ultrasound (DCE-US) analysis, various DCE-US quantitative parameters were obtained, such as the peak enhancement value of the active inflammation bowel wall was statistically higher than that of the normal bowel wall (**d**); Ileocolonoscopy indicated bowel wall thickening, mucosal hyperemia and hyperplasia polyps. The value of the total simple endoscopy score for CD (SES-CD) is 10 (**e**).

**Figure 3 diagnostics-14-00672-f003:**
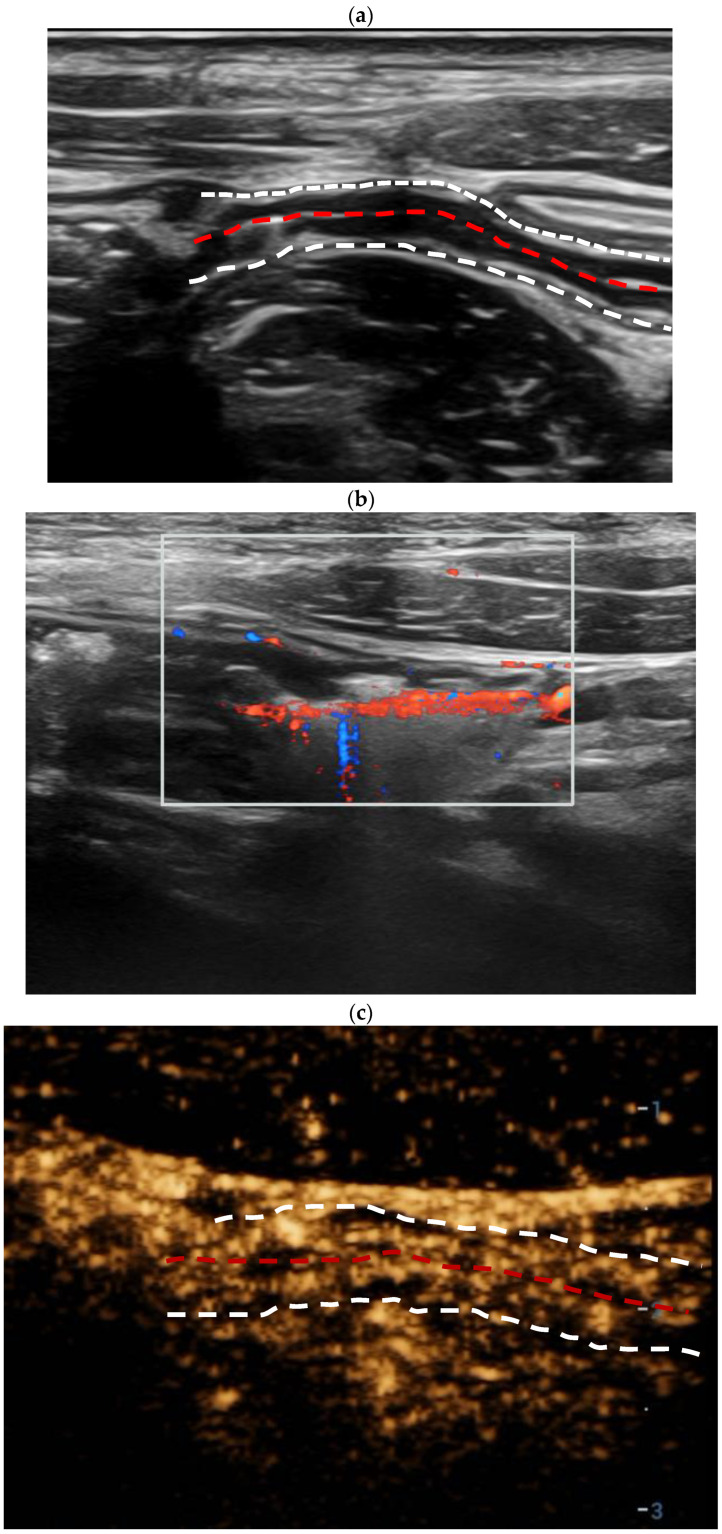
An eighteen-year-old male patient diagnosed with inactive Crohn’s disease. Longitudinal scan of intestinal ultrasound showed the bowel wall layer is distinctively stratified, thickness of the terminal ileum bowel wall is 3.2 mm (the white dotted lines display the boundary between the serosal layer of the bowel wall and the surrounding tissue, and the red dotted lines display the bowel lumen, and the same is applied to the figure legends shown below) (**a**); color Doppler signal of the thickest bowel wall is Limberg II, dot- or short-like blood flow signals could be detected in the thickened intestinal wall (**b**); contrast-enhanced ultrasound (CEUS) of the bowel wall segment with inactive inflammation showed the enhancement of submucosa (enhancement pattern III, (**c**)); two regions of interest (ROIs) were set inside the thickened bowel wall (orange ROI) and in the normal bowel wall (green ROI) as a reference region. The TIC of the thickened bowel wall (orange curve) revealed an oblate curve similar to that of the normal bowel (green curve). After dynamic contrast-enhanced ultrasound (DCE-US) analysis, various DCE-US quantitative parameters were obtained. For example, the peak enhancement value of the active inflammation bowel wall was slightly higher than that of the normal bowel wall (**d**); there were no obvious abnormalities in the intestinal segment on ileocolonoscopy. The value of the total simple endoscopy score for CD (SES-CD) is 2 (**e**).

**Figure 4 diagnostics-14-00672-f004:**
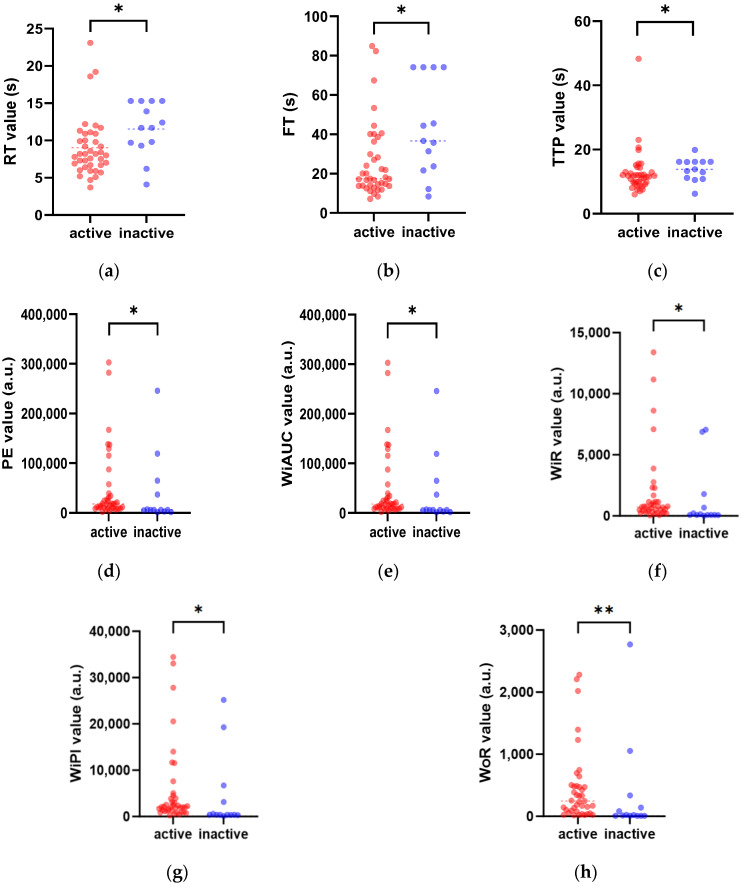
Comparison of the DCE-US quantitative parameters between active and inactive CD groups. The red scatter plots represent patients in active CD, while patients in inactive CD are expressed in the blue scatter plots. The figures display significant differences between the two groups for the perfusion parameters rise time (RT) (**a**), fall time (FT) (**b**), time to peak (TTP) (**c**), peak enhancement (PE) (**d**), wash-in area under the curve (WiAUC) (**e**), wash-in rate (WiR) (**f**), wash-in perfusion index (WiPI) (**g**), and wash-out rate (WoR) (**h**) (* *p* < 0.05, ** *p* < 0.01).

**Figure 5 diagnostics-14-00672-f005:**
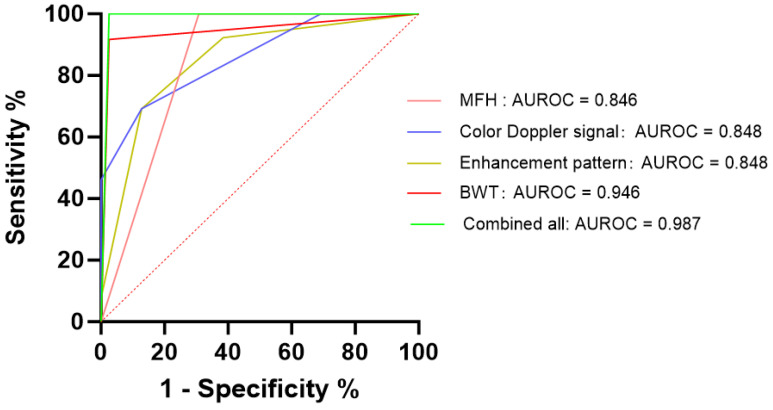
Receiver operating characteristic (ROC) curves of positive intestinal ultrasound parameters and contrast-enhanced ultrasound (CEUS) perfusion quantitative parameters. The ROC curve analysis was compared among the parameters including BWT (bowel wall thickness), MFH (mesenteric fat hypertrophy), color Doppler signal, CEUS enhancement pattern, and a combination thereof (combining all the positive intestinal ultrasound parameters and perfusion quantitative parameters). The area under the ROC curve (AUROC) of combined all was 0.987 (95% CI: 0.958–1.000), with 97.4% sensitivity, 100% specificity, and 98.1% accuracy.

**Table 1 diagnostics-14-00672-t001:** Base characteristics of Crohn’s disease patients.

	Active CD Group (n = 39)	Inactive CD Group (n = 13)	*p* Value
Sex			0.311
Female	14 (35.9%)	3 (23.1%)	
Male	25 (64.1%)	10 (76.9%)	
Age (y)			0.841
Mean ± SD	33.5 ± 13.6	33.3 ± 16.3	
Range	18–60	18–67	
BMI (kg/m^2^)			0.734
Mean ± SD	20.9 ± 3.5	21.3 ± 4.8	
Range	14.2–31.1	16.2–31.8	
CRP (mg/L)			0.001 *
Mean ± SD	32.7 ± 47.5	3.4 ± 7.1	
Range	1–200	1–27	
ESR (mm/h)			<0.001 *
Mean ± SD	30.9 ± 26.1	8.3 ± 7.5	
Range	2–107	2–28	
Location			0.166
Ileal	5 (12.8%)	5 (38.4%)	
Ileocecus	8 (20.5%)	2 (15.4%)	
Colonic	11 (28.2%)	4 (30.8%)	
Ileocolonic	15 (38.5%)	2 (15.4%)	

SD: standard deviation; BMI: body mass index; CRP: C-reactive protein; ESR: erythrocyte sedimentation rate; Location: location of diseased bowel segment, *: *p* < 0.05.

**Table 2 diagnostics-14-00672-t002:** Comparison of intestinal ultrasound features between two groups.

	Active CD Group (n = 39)	Inactive CD Group (n = 13)	*p* Value
BWT (mm)			<0.001 *
Mean ± SD	6.5 ± 1.7	3.2 ± 1.0	
Median (min, max)	6.2 (4.2, 12.6)	3.1 (1.9, 5.3)	
MLN			0.168
Normal	19 (48.7%)	9 (69.2%)	
Enlarged	20 (51.3%)	4 (30.8%)	
MFH			
Absent	12 (30.8%)	13 (100%)	<0.001 *
Present	27 (69.2%)	0	
Color Doppler signal			<0.001 *
Limberg 0	0	6 (46.1%)	
Limberg I	5 (12.8%)	3 (23.1%)	
Limberg II	22 (56.4%)	4 (30.8%)	
Limberg III	7 (18.0%)	0	
Limberg IV	5 (12.8%)	0	
CEUS enhancement			<0.001 *
Pattern I	24 (61.5%)	1 (7.7%)	
Pattern II	10 (25.6%)	3 (23.1%)	
Pattern III	5 (12.8%)	8 (61.5%)	
Pattern IV	0	1 (7.7%)	
Complications			0.028 *
Absent	28 (71.8%)	13 (100%)	
Present	11 (28.2%)	0	

BWT: bowel wall thickness; MLN: mesenteric lymph node; MFH: mesenteric fat hypertrophy; color Doppler signal was classified according to Limberg score [20]; CEUS enhancement patterns were classified in four categories [21]; *: *p* < 0.05.

## Data Availability

Data available on request due to ethical restrictions. The data presented in this study are available on request from the corresponding author. The data are not publicly available due to ethical restrictions.

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
