# Peer review of "Application of Dynamic Contrast-Enhanced Ultrasound in Evaluation the Activity of Crohn’s Disease"

_diagnostics, 2024, doi:10.3390/diagnostics14070672_

Round 1

Reviewer 1 Report

Comments and Suggestions for Authors

Introduction: line 67-  should read fibrotic "stricture"

Otherwise well written and well done.

Comments on the Quality of English Language

Minor revision needed for slight grammatical issues.

Author Response

1. line 67-  should read fibrotic "stricture"

We changed in our revised manuscript accordingly.

Reviewer 2 Report

Comments and Suggestions for Authors

Dear authors, the paper is well constructed and results are validated.

Material and methods are well organized and analysis and results are extensively explored. My only concern is about English, I pointed up just some phrases, but English need to be extensively revised. The images and staystical graph are eccellent. 

Page 1 

line 32 change infected with inflamed

line 38 SES-CD score, the first time you use acronyms please specify

Page 2

Line 64 it specify CD

Line 64-65 maybe the word “ultrasound “ is missing ? And the period is not clear, please rewrite.

Maybe you intend:

According to the 2018 ….. and the European Society of Gastrointestinal and Abdominal Radiology) guidelines, CD can be classified as  active or inactive on the basis clinical, endoscopic, or biochemical value and ultrasound color Doppler signal.

Line 67-69 the period is not clear please rewrite. Maybe you intend:

A non-invasive and dynamic assessment of CD activity is essential for and accurate clinical treatment and follow up.

?

Line 73-74

Subject missing 

And they are (they are both?) commonly used…

Line 77-78

English and

I would detail the findings od SES-CD.

Line 79-80 no full stop and those instead “and”

Line 87

Delay meanwhile and not “had”, report sen spec and accuracy of all the imaging methods you wrote about (CTE is missing) , and English.

Figure 1

Please do not overlap the images

Comments on the Quality of English Language

English should be revised

Author Response

1. Page1 line 32 change infected with inflamed.

We changed in our revised manuscript accordingly.

2. line 38 SES-CD score, the first time you use acronyms please specify

We stated the SES-CD in Line 33 accordingly.

3. Page 2Line 64 it specify CD. Line 64-65 maybe the word “ultrasound “ is missing ?   And the period is not clear, please rewrite. Maybe you intend: According to the 2018 ….. and the European Society of Gastrointestinal and Abdominal Radiology) guidelines, CD can be classified as  active or inactive on the basis clinical, endoscopic, or biochemical value and ultrasound color Doppler signal.

We changed line 64 “color Doppler signal” as “CD” in our revised manuscript accordingly.

4. Line 67-69 the period is not clear,please rewrite. Maybe you intend: A  

non-invasive and dynamic assessment of CD activity is essential for accurate clinical treatment and follow up.

We changed it in our revised manuscript accordingly.

5. Line 73-74Subject missing. And they are (they are both?) commonly used…

Line 74: We added “of CD” followed by “the inflammation activity” in our revisedmanuscript accordingly.

6. Line 77-78I would detail the findings od SES-CD.

We added details in our revised manuscript accordingly.

7. Line 79-80 no full stop and those instead “and”

We changed in our revised manuscript accordingly.

8. Line 87Delay meanwhile and not “had”, report sen spec and accuracy of all the  imaging methods you wrote about (CTE is missing) , and English.

We changed the sentence in line 87 in our revised manuscript accordingly. We added reported sensitivity and specificity of CTE in our revised manuscript accordingly.

9. Figure 1. Please do not overlap the images.

We adjusted figure 1 in our revised manuscript accordingly.

Reviewer 3 Report

Comments and Suggestions for Authors

Title: Application of dynamic contrast enhanced ultrasound in evaluation the
activity of Crohn’s disease

Thank you for inviting me to review this paper.

There are to many abbreviations, that it is difficult to read the paper.

Abstract:

Consider including a sentence about why this study is important e.g. in an introduction.

Introduction

Normally you would write e.g European Crohn’s and Colitis Organization and the European Society of Gastrointestinal and Abdominal Radiology) guidelines (ECCO-ESGAR).

 Could be nice to have some background information about the problem.

Aim – ok.

Methods

Is there an approval number.

When as the study approved (month and year).

How many patients did not meet the inclusion criteria?

There is an overlap between figure 1a and figure 1b

Is figure 1 b investigation the colon? Consider rephrasing “ Schematic representation of

comprehensive ultrasound scan procedure for the colon.”

Who performed the analysis? “. All ultrasound images were digitally recorded in DICOM (Digital  Imaging and Communications in Medicine) format for further analysis”

Please consider including a scopy image.

Results

Table 2, consider including what pattern 1, 2 3 and 4 is.

Please include arrows in Figure 2. Figure 2 d & Figure 3 D is too small. It is too difficult to see Figure 2D/ Figure 3D.

There seems to be 2 Fig 4 – page 11?

Conclusion – the conclusion does not replicate the aim. Please adjust the conclusion.

Comments on the Quality of English Language

ok English 

Author Response

1. Abstract: Consider including a sentence about why this study is important e.g. in an introduction.

We added the importance of this study in abstract in our revised manuscript accordingly.

2. Introduction: Normally you would write e.g European Crohn’s and Colitis

Organization and the European Society of Gastrointestinal and Abdominal Radiology) guidelines (ECCO-ESGAR).

We changed it in our revised manuscript accordingly.

3. Methods: Is there an approval number. When as the study approved (month and  year).

We added approval code (XHEC-C-2023-028-1) in our revised manuscript accordingly. The approval date of the study was March 2023.

4. How many patients did not meet the inclusion criteria?

24 patients were excluded according to the inclusion and exclusion criteria of the study.

5. There is an overlap between figure 1a and figure 1b. Is figure 1 b investigation the colon? Consider rephrasing “ Schematic representation of comprehensive  ultrasound scan procedure for the colon.”

 We adjusted Figure 1 in our revised manuscript accordingly. Figure 1 b represents the comprehensive ultrasound scan procedure for the small intestinal.

6. Who performed the analysis? “. All ultrasound images were digitally recorded in DICOM (Digital  Imaging and Communications in Medicine) format for further analysis”

All ultrasound images were reviewed by two independent senior radiologists, who were blinded to the clinical histories, other imaging findings and histopathological results. Disagreement was settled by consensus.

7. Please consider including a scopy image.

We added scopy images in figure 2 and figure 3 in our revised manuscript accordingly.

8. Results: Table 2, consider including what pattern 1, 2 3 and 4 is.

We added enhancement pattern of CEUS in Table 2 note in our revised manuscript accordingly.

9. Please include arrows in Figure 2. Figure 2 d & Figure 3 D is too small. It is too difficult to see Figure 2D/ Figure 3D.

We added the dotted lines to display the boundary of the bowel wall and the bowel lumen. Figure 2 d and Figure 3 d were enlarged in our revised manuscript accordingly.

10. There seems to be 2 Fig 4 – page 11?

 We changed “Fig 4” to “Fig 5” in page 11 in our revised manuscript accordingly.

11. Conclusion – the conclusion does not replicate the aim. Please adjust the conclusion.

We adjusted the conclusion in our revised manuscript accordingly.
